# Mid-infrared coincidence measurements on twin photons at room temperature

M. Mancinelli[1], A. Trenti[1], S. Piccione[1], G. Fontana[1], J.S. Dam[2,†], P. Tidemand-Lichtenberg[2], C. Pedersen[2] & L. Pavesi[1]

Quantum measurements using single-photon detectors are opening interesting new perspectives in diverse fields such as remote sensing, quantum cryptography and quantum computing. A particularly demanding class of applications relies on the simultaneous detection of correlated single photons. In the visible and near infrared wavelength ranges suitable single-photon detectors do exist. However, low detector quantum efficiency or excessive noise has hampered their mid-infrared (MIR) counterpart. Fast and highly efficient single-photon detectors are thus highly sought after for MIR applications. Here we pave the way to quantum measurements in the MIR by the demonstration of a room temperature coincidence measurement with non-degenerate twin photons at about 3.1 μm. The experiment is based on the spectral translation of MIR radiation into the visible region, by means of efficient up-converter modules. The up-converted pairs are then detected with low-noise silicon avalanche photodiodes without the need for cryogenic cooling.

[1] Nanoscience Laboratory, Department of Physics, University of Trento, Via Sommarive 14, Trento 38123, Italy. [2] DTU Fotonik, Department of Photonics Engineering, Technical University of Denmark, Roskilde 4000, Denmark. † Present address: IRSee ApS, Frederiksborgvej 399, 4000 Roskilde, Denmark. Correspondence and requests for materials should be addressed to A.T. (email: alessandro.trenti@unitn.it).

The mid-infrared (MIR) spectral region (2.5–25 μm) is of interest to many areas of science and technology for mainly three reasons[1]. First, the MIR wavelength range allows for direct probing of the fundamental absorption bands of nearly all gas molecules[2]. For instance, the fundamental absorption band of $CO_2$ in the MIR is roughly two orders of magnitude stronger than its overtone in the near infrared region. The strong and molecule specific MIR light-matter interaction enables sensors with improved sensitivity and specificity forming the basis of vibrational spectroscopy. Second, the MIR wavelength range contains atmospheric windows relevant for free space communication[3] and environmental monitoring[4,5]. Third, room temperature objects emit light at MIR wavelengths, promoting novel thermal camera applications[6,7]. Thus, the unique and attractive characteristics of the MIR range call for enabling techniques to yield sensitive detection. These are already available for the near infrared range (0.7–2.5 μm), where manipulation and detection of photons down to the level of single-photon measurements are becoming mature technological tools. Furthermore, single-photon sensitive measurements allows experiments in quantum optics with unprecedented measurement precision[8]. Near infrared quantum optics has been propelled by the needs of the quantum information science[9,10], where coincidence detection of photon pairs plays a central role. Relevant developments include the characterization of quantum sources of light and of intersubband transitions[11], the development of heralded single-photon sources for a variety of quantum information technologies[8,12] and for ghost imaging[13,14]. One of the key experiments to determine the quantum properties of light is the Bell state measurement, which relies, on photon coincidence measurements. In coincidence detection, the quantum efficiency (QE) and noise affect the signal-to-noise ratio in a quadratic rather than in a linear fashion, thus putting strict requirements to the detector efficiency and sensitivity[15,16]. Nowadays, there is a significant impulse to move quantum optics to the MIR due to the possibilities opened by the MIR for quantum sensing, quantum communication and quantum imaging. Much effort has been devoted to the development of new light sources, leading to many novel technologies such as quantum cascade lasers[17], laser frequency combs[18] and high-power fibre-based sources[19]. In contrast, MIR detectors still face many limitations, mainly due to the inherent sensitivity to unwanted incident black-body radiation and dark current, induced by the finite temperature of the detector itself. Cooling the detector generally reduces these two noise sources; however, it results in costly and bulky devices. Moreover, even when cooled, such MIR detectors generally exhibit poor signal-to-noise ratios compared to near infrared and

visible light detectors[20]. One approach to increase sensitivity is to use superconducting nanowire single-photon detectors (SNSPDs), which offer single-photon sensitivity from visible to near infrared wavelengths, low dark counts, short recovery times, and low timing jitter[21–23]. The SNSPDs system detection efficiency at telecom wavelengths (around 1.55 μm) can be as high as 93%, with a system dark count rate of about 1,000 cps (counts per second)[24]. Reducing the nanowire width the sensitivity can be extended into the MIR. However, the system detection efficiency drops to a value of about 2%, while the dark count rate increases to a value larger than 10 kcps[15]. Therefore, it is clear that an efficient direct MIR single-photon detection is not feasible, either with semiconductor based detector or superconductor technology.

Here we propose to approach this problem by using the concept of spectral translation. The idea is to up-convert the infrared radiation into the visible domain by means of optical nonlinear effects[25]. In this way, the incoming signal is wavelength-shifted to a wavelength interval where efficient and low-noise detectors are readily available and where the influence of unwanted incident room temperature radiation (Planck radiation) is strongly reduced. This enables us to demonstrate coincidence measurements on correlated photon pairs at 3.1 μm.

## Results

**MIR single-photons up-conversion.** The concept of up-conversion is not new, but due to the requirement of high power pump lasers to obtain high conversion efficiency, its use for many practical applications is hampered if not impossible[26]. Recently, it has been demonstrated that the efficiency of the up-conversion process can be dramatically enhanced by placing the nonlinear crystal inside a resonant cavity. A measured QE of 20% has been achieved, due to the high circulating power within the cavity resulting from the low-loss cavity design[27]. This is at least three orders of magnitude higher than the one reported in similar up-conversion schemes[28].

Here, we combine the up-converter module with a state-of-the-art silicon single photon avalanche photodiode (SPAD) to demonstrate single-photon counting in the MIR for quantum optics applications. Compared to the SNSPDs (Table 1), the QE of this MIR single-photon detector is three times higher and the response time and timing jitter is primarily determined by the electronics of the silicon SPAD and not by the up-conversion process itself, which can be considered instantaneous at most time scales employed. In addition, the detector is thermoelectrically cooled instead of the cryo-cooling required for SNSPDs. Note that the thermoelectrically cooling is not an intrinsic

**Table 1 | Comparison between the two MIR single photon-photon counting technologies.**

| | Quantum Efficiency | Coupling losses | System detection efficiency | System dark count rate (at maximum efficiency) | Dead time (ns) | Working temperature (K) | Reference |
|---|---|---|---|---|---|---|---|
| Module + SPAD | 6.5%* | 90% filters 6.1% detector overlap | 0.35% | 1 kcps (at T = 300 K) | 8 | 300 | This work |
| Superconducting Nanowire (SNSPDs) | 2% | Negligible | 2% | >10 kcps† (at T = 300 K) | <10 | 1.5 | 15 |

First row reports the characteristics of the detection system described in this work, while the second row reports the characteristics of the detection system based on superconducting nanowire described in ref. 15. The second column refers to the detector quantum efficiency, the third column to the coupling losses defined as the losses from the input to the detector, the fourth column gives the system detection efficiency computed as the product of the quantum efficiency and the coupling losses, the fifth column gives the system dark count rate at the maximum efficiency, the sixth column the dead time, the seventh column the working temperature.
*The Quantum efficiency of the module + SPAD is given as the product between the up-conversion efficiency of the module (10%) and the SPAD quantum efficiency (65%).
†The Dark Count Rate at T ∼ 1.5 K is 100 cps. This measure was done by blocking the optical coupling between the input optical fibres and the devices with a metal shutter kept at T ∼ 1.5 K. Since black body radiation is a main source of noise in the MIR, we reported in the table the dark count rate measured with a room temperature shutter for the sake of comparison of the two technologies. The reported value without the cold shutter is the fairest value for this comparison.

limitation of the proposed system since there exist SPADs which even operate without temperature control. Another feature that characterizes the proposed system is that it can be tuned to any wavelength within the transparency range of the nonlinear material, provided that phase matching can be achieved. Furthermore, the up-conversion process inherently acts as a spectral and spatial filter to incoming light radiation. In the presented configuration, the spectral and angular acceptance bandwidths are 10 nm and 1°, respectively, at a signal wavelength of 3 μm. This is in strong contrast to direct MIR detection systems. Consequently, as presented in the following sections, it allows both spectral and spatial resolved measurements at the single-photon level.

On the other hand, the fact that a detector has a limited bandwidth might also have drawbacks. Working with a broadband MIR source, it is not possible to get the entire source spectrum in a single measurement. In such cases, it would require scanning of the phase-match condition in order to reconstruct the entire source spectrum.

To demonstrate the capability of the detection unit, we measure the time correlation of MIR photon pairs by means of coincidence measurements. Coincidence counting generally involves two or more detectors connected to an electronic coincidence circuit. When a detector measures a photon, it triggers a signal in the form of an electrical pulse. The temporal coincidence between signals from two separate photon detectors will reveal if there is a correlation in the arrival time of the photons at the detectors.

The twin photons are generated in a free space set-up through spontaneous parametric down conversion (SPDC) in a periodically poled Lithium Niobate (PPLN) crystal and detected by two uncorrelated detection units (Fig. 1a). The essential feature of SPDC is that a single pump photon passing through a nonlinear optical material can spontaneously decay into two daughter photons (signal and idler photons). Energy and momentum are conserved in the SPDC process and, consequently, the signal and idler photons are correlated in momentum, energy and time. In general, the degree of correlation depends on the parameters of the pump, the crystal and the collection optics[29]. Considering the case of three collinear plane waves propagating in a periodically poled nonlinear material along the $z$-axis, such that $\mathbf{k} = k\,\hat{\mathbf{z}}$, the momentum relation is

$$2\pi\frac{n(\lambda_p,T)}{\lambda_p} = 2\pi\left(\frac{n(\lambda_s,T)}{\lambda_s} + \frac{n(\lambda_i,T)}{\lambda_i} + \frac{1}{\Lambda}\right) \qquad (1)$$

where $n(\lambda,T)$ is the refractive index of the material[30] that depends on the wavelength $\lambda$ of the optical signal and on the temperature $T$ of the nonlinear material. $\Lambda$ is the poling period, which is an imposed periodicity on the nonlinear coefficient of the nonlinear crystal. The poling counter-acts the dispersion between the interacting wavelengths, that is, equation (1) is satisfied thanks to the last term that is inversely proportional to $\Lambda$. This technique is called quasi-phase-matching[31]. In PPLN, quasi-phase-matching allows the reaching of high conversion efficiency[27]. Knowing the poling period, it is possible to calculate the spectrum of the SPDC generated photon pairs depending on the crystal temperature (Fig. 2b). The calculation assumes collinear interaction using pump photons at 1.55 μm in a 10 mm long PPLN crystal with a poling period of 34.48 μm (ref. 32). The phase-match condition is temperature-dependent. In particular, the degenerate process, where the signal and idler photons have the same wavelength of 3.1 μm, is achieved at a nonlinear crystal temperature of 135 °C (solid blue curve in Fig. 2b). Tuning the PPLN crystal temperature to 68 °C, two photons with different energies are generated. The generated spectrum contains two side lobes, symmetrically distributed in energy around $\omega_{\mathrm{deg}} = (2\pi c)/\lambda_{\mathrm{deg}}$, where $\lambda_{\mathrm{deg}} = 3.1\,\mu\mathrm{m}$ (solid red curve Fig. 2b). The symmetry is due to the energy conservation: if one photon of the correlated pair is red-shifted with respect to $\lambda_{\mathrm{deg}}$, the other photon must be blue-shifted by the same amount . The spectral bandwidths of the two SPDC peaks, defined as their FWHM, are 50 nm (58 cm$^{-1}$) for the 2.89 μm peak and 66 nm (58 cm$^{-1}$) for the 3.34 μm peak, respectively. In the case of degenerate SPDC, the bandwidth is increased to 200 nm (208 cm$^{-1}$).

Another second-order nonlinear process is employed to detect the SPDC generated photons: frequency up-conversion via sum frequency generation (SFG). Two detection units, composed by an up-converter module (Fig. 1b) and a Si SPAD, are used. The up-converter modules are also based on PPLN crystals (20 or 5 mm long), but with different poling periods than the SPDC crystal, in order to phase-match the up-conversion processes with the intra-cavity pump laser. In the SFG process, the incoming MIR photon is mixed with a pump photon at 1,064 nm to generate a photon in the visible range. The QE of the SFG process is boosted by the 100 W of continuous-wave circulating pump power, present within the cavity of each of the up-converter modules. The generation and detection stages are sketched in Fig. 2a.

The bandwidths of the two up-converter modules can be compared to the SPDC spectra (Fig. 2b). It is possible to fine-tune the phase-matched wavelengths of the modules optimizing the temperatures of the PPLN crystals (Fig. 3a) or by changing the relative angle between the incoming MIR light and the pump beams in the modules[27]. Each module has five poling periods ranging from 21 to 23 μm in steps of 0.5 μm. In order to phase-match the degenerate SPDC photons, the two up-conversion modules have been set to $T_{\mathrm{deg}} = 145\,°\mathrm{C}$ using a poling period of 21.5 μm (solid green curve in Fig. 2b). The spectral acceptance bandwidth of the two modules is 7 nm, using 20 mm long PPLN crystals. Figure 3b shows a demonstration of the module filtering capability at 3.1 mm, ($T_{\mathrm{deg}} = 145\,°\mathrm{C}$, 21.5 mm poling period, 5 mm long). It matches well with the theoretically predicted acceptance bandwidth of 26 nm (using a 5 mm long PPLN crystal at 3.1 μm). This is expected, as the bandwidth scales inversely proportional to the crystal length. The measured spectral response for the 20 mm long crystal is reported in Supplementary Fig. 1b. Fig. 3c shows the SPDC signals while temperature scanning the SPDC crystal and keeping the detector parameters constant. The module settings are the same of Fig. 3b. In Fig. 3d the experimental SPDC spectrum is reported. The data were collected as a function of the SPDC crystal temperature, in order to show the degenerate and non-degenerate processes. To cover a wide spectral region, we used three different poling periods in the detector: (1) 21 μm, (2) 21.5 μm and (3) 22 μm. For each poling period, we scan the phase-matched wavelength with the SFG crystal temperature, according to the experimental characterization reported in Fig. 3a.

For correlation measurements and in case of non-degenerate SPDC emission, one module is tuned to phase-match the signal wavelength at 2.89 μm using a poling period of 21 μm and a crystal temperature of 54 °C (dashed black curve in Fig. 2b), while the second module has to phase-match the idler photon at a wavelength of 3.34 μm, using a poling period of 22.5 μm and a PPLN temperature of 53 °C (solid black curve in Fig. 2b). The bandwidths of the up-conversion modules are 5 nm (6 cm$^{-1}$) for the shorter 2.89 μm wavelength, and 10 nm (9 cm$^{-1}$) for the longer 3.34 μm wavelength.

Comparing the SPDC bandwidths with the bandwidths of the up-conversion modules, it is clear that the detection system is more efficient for the non-degenerate process than for the degenerate case (Fig. 2b). Considering the spectral overlap of

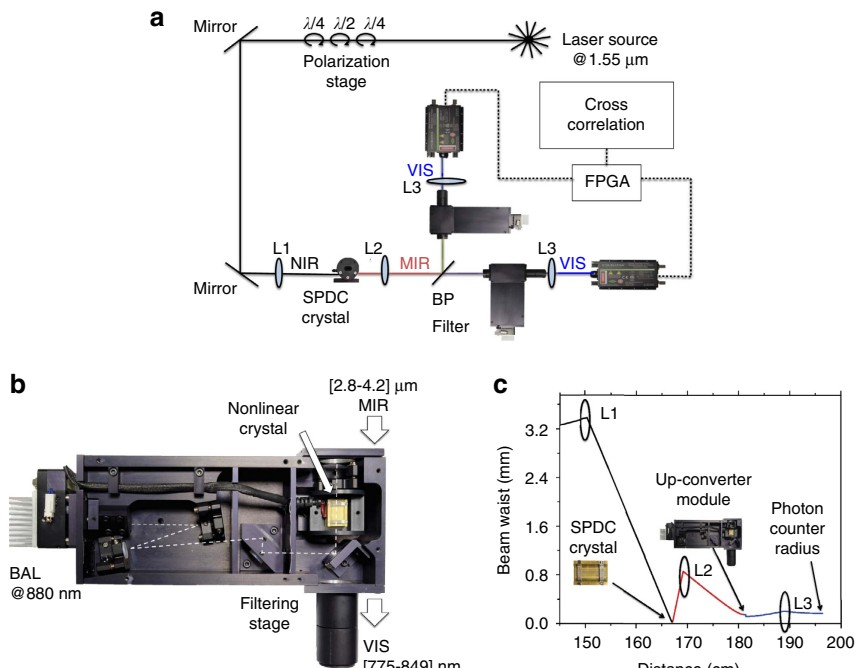

**Figure 1 | Experimental set-up.** (**a**) Sketch of the experimental set-up. A continuous-wave infrared laser, amplified by an erbium-doped fibre amplifier, pumps a PPLN crystal. The polarization of the pump light is aligned with the optical axis of the PPLN crystal by a polarization stage. Lens L1 has a focal length of 150 mm and focuses the beam, whose waist before L1 is 2.25 mm, to a beam waist of 35 μm in the PPLN crystal. The PPLN crystal contains 16 different poled regions each with a transverse cross section of 500 μm by 500 μm. The nominal poling period used is 34.48 μm. The fine-tuning of the phase-match condition is performed by changing the temperature of the PPLN crystals, which are mounted in ovens, by temperature controllers. The generated MIR photon pairs and the residual 1.55 μm pump are collimated by a CaF$_2$ converging lens L2, whose focal length is 18 mm. The generated photon-pairs are split by an optical bandpass filter centred at 3 μm, with a bandwidth of 500 nm. The idler wavelength at 3.343 μm is reflected and sent to one up-conversion detection module, while the signal photon at 2.89 μm is transmitted through the bandpass filter to the second up-conversion detection module. Residual pump photons at 1.55 μm are eliminated by Germanium windows, placed at the entrance of each of the up-conversion modules. After the up-conversion process, the generated visible light is collimated and focused to match the small sensitive area of the photon counter by lens L3. L3 has a focal length of 75 mm and it is placed at a distance of 73.8 mm from the photon counter. The up-converted beam waist is about 300 μm. (**b**) Photograph of the up-converter module. The description of the different parts is reported in the Methods section. (**c**) Simulation of the beam waist as a function of the distance along the optical axis of the system from the laser to the silicon-based SPAD detector. The colours refer to: pump beam at 1.55 μm (black), SPDC-generated beam at 3.1 μm (red), up-converted beam 792 nm (blue).

the degenerate SPDC spectrum and of the module bandwidth, each module will only be able to up-convert about 3% of the emitted photons. In the non-degenerate case, we estimated 11% up-conversion efficiency from the signal band, centred at 2.89 μm, and 16% up-conversion efficiency from the idler band, centred at 3.34 μm.

The difference in the up-conversion efficiency between the degenerate and non-degenerate case is the main reason to exploit the non-degenerate SPDC process. In addition, there are other experimental issues favoring the non-degenerate process, namely the ease of separating the signal and idler photons by a simple dichroic filter, and the lower temperature in the PPLN of the up-converter modules, which reduces the noise due to black-body radiation.

**Coincidence measurement.** To validate the model experimentally we implemented a free-space optical set-up (Fig. 1a). A laser pumps the first nonlinear crystal to generate time correlated SPDC MIR photon-pairs that are spectrally translated into the visible spectrum by two up-converter modules and detected by two Si SPADs. The overall efficiency in collecting the generated MIR photons is strictly determined by the dimension of the beam waist. Hence, the various focal lengths and the relative distances between the lenses and the nonlinear crystals have to be carefully selected. For this reason a numerical simulation, based on the

evaluation of the beam along the optical path, was performed in order to achieve the optimum condition (Fig. 1c). A detailed discussion is presented in the Methods section. The spectrum of the generated SPDC photons in the MIR was measured (Fig. 3d). We estimated a total loss of 25 dB from the output facet of the SPDC crystal to the input of the single-photon detectors (Fig. 1a). A detailed analysis of the losses is reported in the Methods section. The output TTL voltage of the Si SPADs is fed into a Field Programmable Gate Array (FPGA) digital correlator that provides the coincidence rate. The FPGA is programmed to make a real-time cross-correlation between the TTL signals of the two Si SPADs. The coincidence window is 1.33 ns.

A clear peak of coincidences in time between the signal and idler photons is observed (Fig. 4b). This coincidence peak demonstrates the time-correlated nature of the MIR generated photon pairs and the capability of our systems to perform this kind of measurement. A schematic of the experimental set-up used to perform coincidence measurement on the photon pairs is reported in Fig. 4a. It is important to note, that the narrow spectral acceptance of the modules reduces the overall number of detected photon pairs. However, if the signal photon is within the bandwidth of the signal detector, the idler photon will also be within the bandwidth of the idler detector—due to energy conservation. Thus, the overall fidelity of pair detection (pairs to background signal) is not compromised by the narrow bandwidth of the detectors, as long as the wavelength bands for the signal

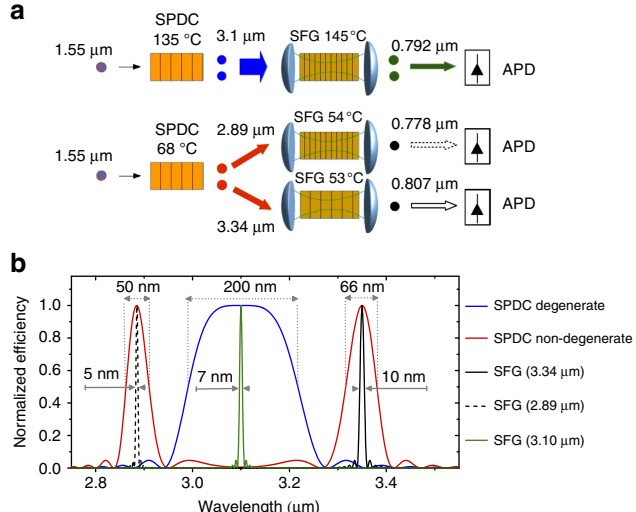

**Figure 2 | Module spectral response.** (**a**) Schemes of the generation and detection stages in the case of degenerate (top) and non-degenerate (bottom) SPDC process. (**b**) Simulated down-conversion spectra and module bandwidth as a function of the temperature. The solid blue curve refers to the degenerate down-conversion process around 3.1 μm, which is achieved at 135 °C. The solid red curve refers to the non-degenerate down-conversion process, which is achieved at 68 °C. They have been derived considering a collinear process, with the pump photons at 1.55 μm and a 10 mm long crystal with a poling period equal to 34.48 μm. The bandwidth of the up-converter module is reported also in the graph.
The details of the up-converter module are reported in the main text.

and idler is carefully aligned to match the generated pair, and has matching bandwidths. It is crucial to reduce the overall detection loss after the generation of SPDC in the PPLN crystal. It is enough to lose just one photon out of the pair to eliminate the coincidence and, consequently, the loss has to be included both the signal and idler simultaneously for successful coincidence measurements. Ideally, the coincidence peak will appear at a zero delay, but due to differences in the propagation delays of the electrical signals between detectors and the correlation unit, it appears at 8 ns (Fig. 4b).

The coincidence background is the rate of coincidences that does not result from the detection of both photons from a single pair. The background coincidences counts are uncorrelated in contrast to the time correlated SPDC photon pairs. The counts from uncorrelated photons that happen to arrive at the same time as the coincidence photons and true coincidence counts are indistinguishable. In the end, the true signal appears as a peak imposed on the background noise signal. The signal-to-noise ratio in coincidence measurements is usually named coincidence to accidental ratio (CAR). CAR is calculated as the number of coincidences within the coincidence window of 1.33 ns, divided by the average of the background counts on the same time window taken apart from the peak[33]. It is enough to integrate for 10 s to have a CAR of 15.7 ± 0.4, with a coincidence rate of 105 ± 1 cps and an accidental mean rate of coincidence of 6.7 ± 0.1 cps. Clearly, the more measurement raw data are integrated, the smaller is the error in the final measure. A computer program was used to simulate the expected coincidence rate as a function of the count rate and dark count rate of each module, taking into account the overall losses of the system (Fig. 4b). The simulated result (solid red curve) is

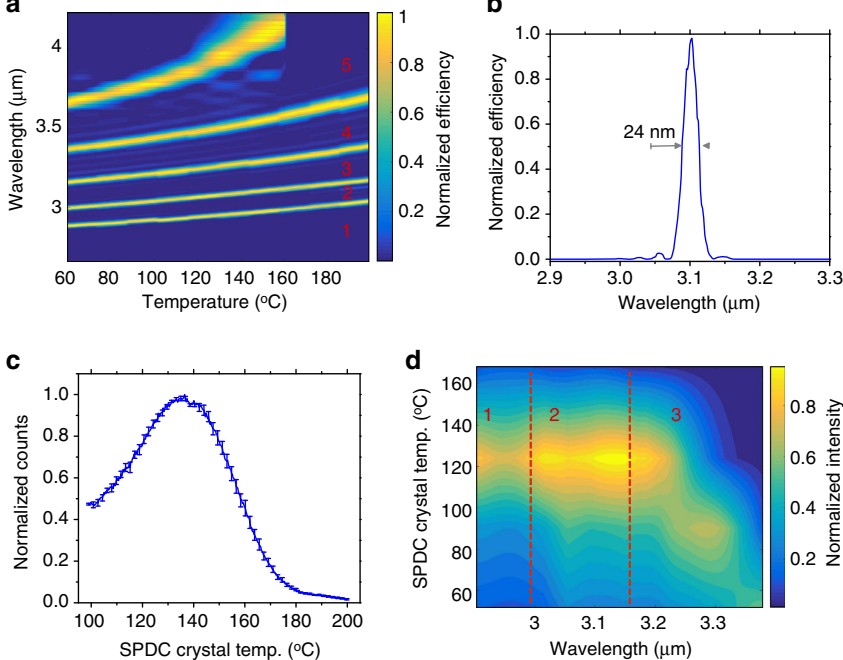

**Figure 3 | Characterization of the up-converter module.** (**a**) Phase matched wavelength and bandwidth for the different poling periods as a function of the temperature. The crystal length is 5 mm. Starting from the bottom, the different poling periods are: (1) 21 μm, (2) 21.5 μm, (3) 22 μm, (4) 22.5 μm and (5) 23 μm. (**b**) Filtering capability of the up-converter module, set to phase match 3.1 μm (21.5 μm poling period, 145 °C crystal temperature).
(**c**) Experimental demonstration of the collected photons of the module at 3.1 μm as a function of the SPDC crystal temperature. The module settings were constant during the measurement (21.5 μm poling period, 145 °C crystal temperature). Error bars: ±1 s.d. due to Poissonian statistics. (**d**) Experimental spectrum of the SPDC nonlinear process as a function of the SPDC crystal temperature measured by the module. Different poling periods (labelled as 1, 2, 3 regions in the figure) of the SFG crystal in the module were used to scan the wavelength. Experimental details about the measurement are reported in the main text. The colour bar represents the efficiency normalized for the maximum value.

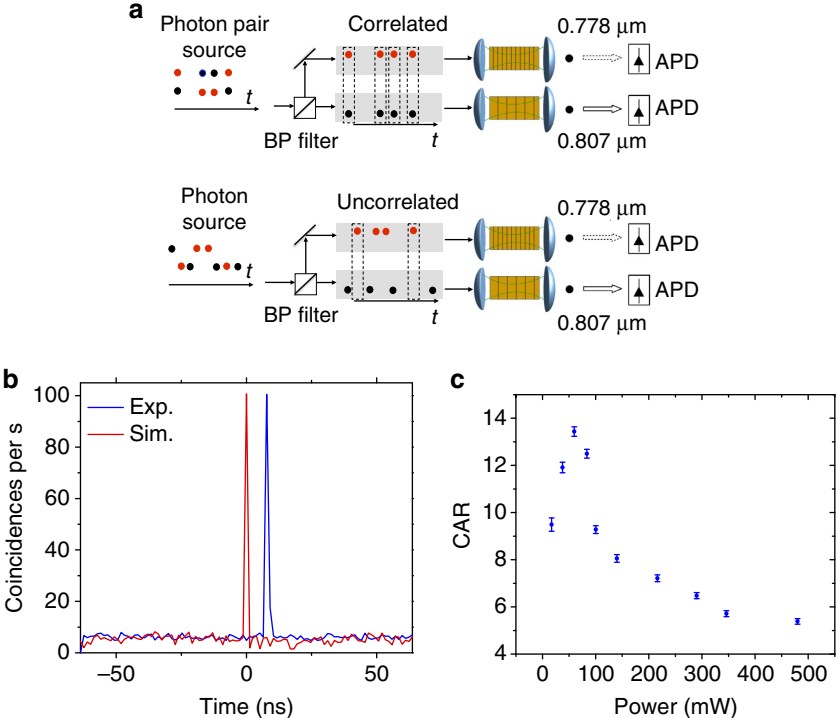

**Figure 4 | Coincidence measurements.** (**a**) Sketch of the experimental set-up used to perform coincidence measurement on the MIR photon pairs, emitted by the PPLN SPDC crystal. (**b**) Coincidence rates, measured (blue line) and simulated (red line) after 10 s of integration with a pump power of 70 mW, in the SPDC crystal (Fig. 1b). The coincidence window is 1.33 ns. The coincidence peak rate is 105 ± 1 cps. (**c**) CAR as a function of the pump power on the SPDC crystal. The data were collected for 50 s of integration time to reduce the errors in the high-power regime. Error bars are derived from the s.d. of the coincidence peak and the s.e. of the mean accidental background rate.

compatible with the experimental data (solid blue curve). The result of the simulation is outlined and commented in Supplementary Note 2.

There is a trade-off in the maximum pump power that can be used for the SPDC process to maximize the CAR. In the low pump power regime, the number of generated photon pairs through the SPDC process increases linearly with the pump power[34]. However, above a certain threshold the generation of more than one pair within the same time-slot will decrease the CAR. This process is called multi-pair emission in SPDC and it is well known to have a detrimental effect on the coincidence measurement[35–37]. An experimental characterization of the CAR as a function of the pump power incident on the SPDC crystal is performed, with an integration time of 50 s, showing that above 70 mW the CAR decreases due to the multi-pair photon emission from the SPDC crystal (Fig. 4c). This measurement is consistent with previous works[38].

The experimental coincidence measurement was performed with an incident pump power set to an optimal value of 70 mW (Fig. 3b). The conversion efficiency of the SPDC process is estimated to be $\eta_{SPDC} = -110\,dB$. After optimization of the temperatures of the PPLN crystals, the count rate for Module1 is 153 kcps (signal photons), with a background rate of 9 kcps. The count rate for Module2 is 150 kcps (idler photons), with a background rate of 14 kcps. The main contribution to the background count rate in both up-converter modules originates from spurious signals at the output of the module itself. A detailed analysis is given in the Methods section.

## Discussion

We have presented a room temperature coincidence measurement of correlated MIR photons. Our detection system outperforms single-photon detectors based on superconducting nanowires in the MIR, which are operated in the few degrees

Kelvin regime, in terms of quantum efficiency, speed and noise. This is a key result because coincidence measurement is the standard technique today for photon pair detection, which is at the core of many quantum optics experiments, and it is essential in quantum communication protocols based on entangled photons[10,39–41]. The proposed system is also a part of a proof of concept towards fully integrated devices. In fact, the integration of an up-converter module on a silicon chip is possible by using silicon SPADs coupled to an up-converter stage realized in the SiN or SiON alloy based on third order nonlinear effects. The results reported here paves the way to quantum optical applications in the MIR, where the ability to perform coincidence measurement is a cornerstone.

## Methods

**The up-converter module.** The construction of the up-converter modules follows[27]. A module comprises of two distinct parts (Fig. 1b). First, the up-conversion crystal which is a 20 mm (5 mm) long PPLN fabricated for sum frequency generation between a 1,064 nm pump laser beam and an incident radiation in the 3 μm wavelength range. The PPLN from Covesion Ltd. has five poling periods ranging from 21 to 23 μm in steps of 0.5 μm. Each poling channel has a 1 mm by 1 mm aperture. In order to reduce reflection losses, the crystal is AR-coated for 1,064 nm and 3 μm radiation, respectively. Using a temperature controller, fine-tuning of the phase-match condition can be achieved. Second, the up-conversion crystal is placed inside a high-finesse 1,064 nm laser cavity in order to enhance the up-conversion probability, that is, the QE. The laser cavity consists of a Nd:YVO₄ laser cavity, optically pumped by a 4 Watt broad area diode laser (BAL) at 880 nm. The 1,064 nm laser cavity includes seven mirrors in order to be configurable for different spot sizes in the laser crystal and the PPLN, respectively, and to further remove residual pump photons in the near infrared from the BAL diode. Up to about 100 W of circulating 1,064 nm laser power can be achieved, providing an increase by about 100 times in QE using a 4 W BAL. A number of filters were included to remove unwanted light from entering the SPADs.

**Experimental set-up.** The experimental set-up used for the coincidence measurement consists of a fibre-coupled, continuous-wave laser with tuning range

from 1.52 to 1.565 µm and maximum output power of 70 mW (Fig. 1a). The laser source is amplified using an erbium-doped fibre amplifier with maximum output power of 800 mW. We controlled the polarization of the light from the laser with a free-space polarization controller stage composed by two quarter-wave plates and one half-wave plate. The pump beam at 1.55 µm is focused by L1 into a PPLN, crystal pumping the SPDC process. It was proved experimentally that the optimum condition is achieved when the Rayleigh range is approximately half the length of the crystal. L1 has a focal length of 15 cm and matches well this condition in our case. The crystal is 10 mm long and contains sixteen periodically poled regions ranging from 24.06 to 36.95 µm, located side by side. Each poling period has a 500 µm by 500 µm aperture. The temperature of the nonlinear crystal is controlled with 0.1 °C accuracy from just above room temperature to 200 °C. It is important to note that the phase-matched wavelengths change with poling period as well as with crystal temperature. A computer program is used to calculate the required poling period and corresponding crystal temperature to obtain phase-matching at a desired wavelength (Fig. 2b). The oven housing with the nonlinear crystal can be translated, allowing for selection of the required poling period. The generated photon-pairs are then split and directed to the two modules, using an IR optical bandpass filter centred at 3 µm, with a bandwidth of 500 nm. The idler wavelength at 3.343 µm is reflected and the signal photon at 2.89 µm is transmitted through the bandpass filter. Residual pump photons at 1.55 µm are eliminated by a Germanium window, placed at the entrance of each of the up-conversion modules. To maximize the efficiency of the up-conversion, it is necessary for the incoming MIR beam to have a beam waist comparable to the beam waist of the laser beam in the up-conversion module. It was found that if the second lens L2 is placed 2 cm from the SPDC crystal and 11 cm from the up-conversion module, the obtained beam waist of the MIR beam, in the SFG crystal, is 194 µm (Fig. 1c). The beam waist of the laser beam in both up-conversion modules is about 200 µm.

The up-converted photons are detected using silicon-based, single-photon counters (SPCM-AQRH *Excelitas*), each coupled to an up-converter module. The SPCM-AQRH is based on a silicon avalanche photodiode with a circular active area, achieving a peak photon detection efficiency of about 65% at 792 nm over a 180 µm diameter with unmatched uniformity over the full active area. A TTL level pulse is generated for each photon detected and the signal is available at the BNC connector at the rear of the module. The detector works in free running mode with a dead time of 20 ns and a linear dynamic range of 36 dB. The typical maximum count rate before 100% saturation is 40 Mcps. Better SPADs are available on the market if improved performance is needed.

**Conversion efficiency and noise characterization.** The total loss from the output facet of the SPDC PPLN to the single-photon detector has been estimated to be about 25 dB. Approximately 10 dB originates from the conversion efficiency of both up-converter modules, which is 10% (Fig. 1c). This value can be increased up to 20% by suitable optimization of the module[27]. L2 is made of CaF$_2$ material, which is transparent at the relevant mid-IR wavelengths and accounts for a loss of 0.45 dB. The filters used after the module account for 0.5 dB. They are short pass (cut-off at 1,000 nm), short pass (cut-off at 850 nm), long pass (cut-off at 600 nm) and long pass (cut-off at 750 nm) wavelength filters. The background count rate of each module is mainly due to spurious signals coming from the module itself (Supplementary Fig. 2). The filter transmittance spectra of both modules are reported in Supplementary Fig. 3. The detection efficiency of the silicon single-photon counter is 65% (1.9 dB). The remaining 12.15 dB losses are due to a non-perfect overlap with the active area of the silicon detector, which has 180 µm diameter. In our implementation, a tighter focus was not feasible, due to intrinsic limiting lens aberrations, amplified at shorter focal lengths.

An experimental characterization of the CAR as a function of the injected power to the BAL, that is, the module intracavity power, is reported in Supplementary Fig. 4.

Looking at Table 1, where the MIR single-photon counting technologies are compared, it seems that the two have comparable performances when we neglect the stringent SNSPDs cryogenic cooling requirement. However, our detection system can be substantially improved. In details, it was demonstrated a high up-conversion efficiency of 20% (ref. 27). Furthermore, commercial silicon-based photon counters show a quantum efficiency of 80% (model ID 120 http://www.idquantique.com), with an active area of 500 µm in diameter. The use of this SPAD, together with an optimized design of the up-converted visible photon waist at the detector, will allow reducing the coupling losses and, eventually, eliminating them. Therefore, in an optimized system, the System Detection Efficiency can be increased up to 15%.

Moreover, the compactness of the system can be further increase by coupling both the input and the output of the up-converter module with optical fibres. This will simplify the coupling of the input signal as well as will drastically reduce the coupling losses to the Si SPAD.

**Data availability.** The authors declare that the data supporting the findings of this study are available from the corresponding authors on request.

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

## Acknowledgements

We would like to thank dott. Massimo Borghi for the helpful discussions. Provincia Autonoma di Trento supported this research by the SiQuro project within the Grandi progetti 2012 call.

## Author contributions

J.S.D., P.T.-L. and C.P. designed and fabricated the module. M.M. tested the module. M.M., A.T., S.P., G.F. mounted the coincidences set-up and performed the measurements. M.M., L.P. and C.P. conceived the experiment and coordinated the work. All the authors wrote the paper.

## Additional information

**Competing interests:** The authors declare no competing financial interests.

**Publisher's note**: 

