## [Peer Review File · Nature Communications]

Reviewers' comments:

Reviewer #1 (Remarks to the Author):

Dear Editor,

please find my report for

Mid-infrared coincidence measurements on twin photons at room temperature
by M. Mancinelli et.al

Let me begin this report with a brief summary of the work. The authors demonstrate a way to achieve mid-infrared (MIR) photon detection at 3.1 μm . They achieve this by spectral translation using up-conversion to the visible spectrum near 800 nm where they can be detected with low-noise silicon avalanche photodiodes. First let me say that single photon detection in the MIR band is an important problem that needs to be solved. Overall this is a nice article but there are a number of issues that need to be addressed before publication can even be considered. The most important is that the authors have been very brief in this article (as it was obviously written for a different journal as it is not in the style of the Nature Communications format). I would like to see the main text explained and more details and discussions included. Further in their reply the authors need to address the following points

- I find the first bold text paragraph quite terse and not generally helpful introducing what they are doing, why it is important and its implications for the future. I believe this paragraph needs to be rewritten. Next in this paragraph, the authors say "Most quantum sensing devices use visible light". I do believe this is true any more - especially for magnetic field sensors. Many of these operate coming outside the visible light spectrum. Further the authors also say "then detected with low-noise silicon avalanche photodiodes with single-photon counting ability and without the necessity of any cooling system". While they may not have chosen to use a cooling system, they are more efficient detectors at this wavelength than silicon avalanche photodiodes that do require active cooling. Use better and more efficient detectors is likely to be an advantage for many quantum applications.
- In the next paragraph "the mid-infrared (MIR) spectral region (2-25 μm) is of interest to many areas of science and technology for mainly three reasons ..." the authors give a short explanation as to why single photon detection is important here. I like this as it helps set the context, but I want to see it explained and more details put in. For example when the authors comment on the "direct probing of the fundamental absorption bands of nearly all gas molecules" please give a specific example the average reader can understand. This would help set the context for this paper and at the same time, the authors can discuss the limitation we currently have from not having good detectors in this regime.
- The paragraph that follows this (starting "Another intense area of research") is not linked well to the previous one. In fact the first three paragraphs (excluding the bolded one) do not flow well together and seem more like a collection of ideas. Integrating them together is essential for a smooth flowing story.
- On page 3 the authors say "One approach to increase sensitivity is to use Superconducting Nanowire Single-Photon Detectors (SNSPDs), which offer single-photon sensitivity from visible to MIR wavelengths, low dark counts, short recovery times, and low timing jitter. Recently, it was suggested to extend the sensitivity of those detectors into the MIR wavelength range by reducing the nanowire width to 30 nm. However, in the MIR the performance of SNSPDs decrease drastically in terms of detection efficiency and recovery time, even when operated at the required few degrees Kelvin." This is a very important paragraph but I would like to see it explained. How bad is the performance going to be at 3.1 μm for instance. Will they be better than the experimental results presented here?

- Next the authors say "This scenario changed when DTU Fotonik demonstrated that the efficiency of the up-conversion process can be dramatically enhanced by placing a nonlinear crystal inside a resonant cavity. A measured quantum efficiency of 20% has been achieved, due to the high circulating power within the cavity resulting from the low-loss cavity design. Here, we couple the up-converter module with a state-of-the-art silicon Single Photon Avalanche Photodiode (SPAD) to demonstrate single-photon counting in the MIR for quantum optics applications". This really makes it sound like the big innovation was from DTU Fotonik and that this work is just a logical extension of it. This seems to reduce the significance and impact of this work and its justification for publication in Nature Communications.

- How efficient are the high-efficient up-converter modules? Are they only 20%. If so, are they really high efficiency?

- Losses are quoted at 25 dB from the output facet of the SPDC crystal to the input of the single-photon detectors. This seems very high when working at the single photon level. Please discuss in more detail and the prospect for future improvements

- In table 1, quantum efficiency are given. The majority of the single photon detection community have moved from quantum efficiency to system detection efficiency. Please use the number for this instead. Next when a quantum efficiency for the module + SPAD is given at $10\% \times 65\% = 6.5\%$ it seems that no account of coupling efficiency between the module and SPAD is considered. It must be included for a fair comparison.

Now let me summarize my opinion. I believe this is an interesting article, however from what is presented, it is not obvious it gives a better performance than other approaches. The authors need to make sure such a comparison is done in a fair way. Given this I can not recommend it for publication in its current form. A major revision is needed.

Reviewer #2 (Remarks to the Author):

This paper is a useful demonstration showing upconversion of MIR twin photons to a wavelength where they can be detected by a silicon photon counter but there are a number of issues that need addressing before publication, most of which have to do with clarity.

Table 1 claims that the overall QE for the detection is $6.5\% = 10\% \text{ upconversion} \times 65\% \text{ SPAD}$ but Methods section says "the total loss from the output facet of the SPDC PPLN to the single photon detector has been estimated to be 25 dB. 10 % comes from the conversion efficiency of both upconverter modules which is 10%." Why is the additional 15 dB not part of the efficiency of the scheme? Something is inconsistent.

The units of most of the count rates mentioned should be "counts per second" not "Hz" which is defined as cycles/second.

In the conclusion it is claimed that a measurement of entangled MIR photons was presented. What was presented was a measurement of correlated photons, not entangled photons. No entanglement was demonstrated.

Reference is made to the "dark count rate in both upconverter modules". What is being referred to is really the "background rate" which is due to an optical process. This is not what is usually meant by the term "dark counts".

A minor point, but it is not quite correct to say that the scheme operates at room temperature, as the SPADs are TE cooled.

In the methods section, it mentions that the PPLN has a number of waveguides with different poling periods, but it is never stated (at least until much later) that just one of those is chosen for the experiment. For clarity, that should be stated explicitly. A simple additional phrase would fix this.

"A number of filters were included to remove unwanted light from entering the SPADs." Filtering in the presence of strong pump light is key to the success of this scheme, so it would be good specifically list the transmittances of those filters.

Fig. 2 caption: It states that "the bandwidth is reported also in the graph." No it is not, but it should be. Also I would use arrows to point from the labels to the curves. As it is, it can be hard to figure out which color is which.

In the supplementary section Fig. S1 has no color scale bar.

In the supplementary section it states "A trade-off has been found in term of signal to noise ratio, at a current value of 3.6 A." It would be useful to show a graph of that.

Figure S3 is really a trivial result that the does not warrant an additional graph especially when Fig. 3b already shows it.

The last page of the supplementary section essentially presents how the efficiencies of each of the two detection channels can be calibrated by a two photon source. This calibration technique has been demonstrated numerous times. There should at least be some reference included to provide the reader a clue to that.

Reviewer #3 (Remarks to the Author):

The authors extend the method of spectral conversion by sum-frequency generation in a nonlinear crystal to the MIR spectral region, which is of high relevance to science and applications. The idea was demonstrated previously in different spectral regions. Still, in my opinion, the current demonstration is relevant technologically.

The manuscript is well-written. The methods section is detailed, which is important for a technical contribution.

It would interesting to see more experimental results in the paper. For example a part of Fig. S1 could be added as an inset to Fig. 1.

An exemplary measurement (single photon interference or Hong-Ou-Mandel effect) performed with the help of the presented detection scheme would add value to the manuscript. Alternatively the authors could demonstrate the filtering capabilities of the scheme and measure the spectrum of SPDC using the relation presented in Fig. S1 c)

Let me go over the text giving more specific remarks. Numbers given enumerate the lines of the manuscript.

75. I understand that it might be advantageous to have spatial and spectral filtering, but it is often a drawback that a detector has a limited bandwidth. We see this problem even in the reported situation (see Fig. 2 b). I suggest to reformulate this part.

88-91. What authors write here is a simplification, which is in general not true. Signal and idler photons don't need to be correlated -- this depends on the parameters of the pump, crystal and collection optics.

92. Formula 1: The authors discuss quasi phase matching (QPM), but use a formula without the contribution proportional to $1/(\text{poling period})$ which is the key point for QPM. This should be corrected.

104. Minor comment: Just to be precise and clear I wouldn't say that the spectrum is symmetrically distributed around λ_{deg} (this would be true for ω_{deg} ...).

175. Which time slot do the authors refer to here? In my understanding time-slot relevant to CAR is the coincidence window width and the time relevant to multi-pair emission is the coherence time of SPDC. This should be clarified.

Additionally, saturation of the detectors could be mentioned as another limitation (I realize that it is due to APDs and not to the conversion, but for the completeness it might be mentioned).

268. Fig. 1. a) could be more schematic -- simplified to increase clarity. d) it is not written in the inset how the data for this plot was measured.

333. Table 1: It would be informative to add the dark count rate of SNSPDs @ 1.5K.

In summary, I think that the manuscript could be accepted after some revisions .

Subject: response letter for manuscript NCOMMS-16-20415

Title: "Mid-infrared coincidence measurements on twin photons at room temperature"

by M. Mancinelli, A. Trenti, S. Piccione, G. Fontana, J. S. Dam, P. Tidemand-Lichtenberg, C. Pedersen and L. Pavesi.

In the following, we answer and discuss point by point the comments carried out by the reviewers.

Legend:

Qx: Reviewer's comment/question number "x".

Ax: Answer and discussion to the reviewer's comment/question Qx.

Reviewer #1 (Remarks to the Author):

Dear Editor,

please find my report for

**Mid-infrared coincidence measurements on twin photons at room temperature
by M. Mancinelli et.al**

Let me begin this report with a brief summary of the work. The authors demonstrate a way to achieve mid-infrared (MIR) photon detection at 3.1 μm . They achieve this by spectral translation using up-conversion to the visible spectrum near 800 nm where they can be detected with low-noise silicon avalanche photodiodes. First let me say that single photon detection in the MIR band is an important problem that needs to be solved. Overall this is a nice article but there are a number of issues that need to be addressed before publication can even be considered. The most important is that the authors have been very brief in this article (as it was obviously written for a different journal as it is not in the style of the Nature Communications format). I would like to see the main text explained and more details and discussions included. Further in their reply the authors need to address the following points

- I find the first bold text paragraph quite terse and not generally helpful introducing what they are do, why it is important and its implications for the future. I believe this paragraph needs to be rewritten. Next in this paragraph, the authors say "Most quantum sensing devices use visible light". I do believe this is true any more - especially for magnetic field sensors. Many of these operate coming outside the visible light spectrum. Further the authors also say "then detected with low-noise silicon avalanche photodiodes with single-photon counting ability and without the necessity of any cooling system". While they may not have chosen to use a cooling system, their are more efficient detectors at this wavelength than silicon avalanche photodiodes that do require active cooling. Use better and more efficient detectors is likely to be an advantage for many quantum applications.

- In the next paragraph "the mid-infrared (MIR) spectral region (2-25 μm) is of interest to many areas of science and technology for mainly three reasons ..." the authors give a short explanation as to why single photon detection is important here. I like this as it helps set the context, but I want to see it explained and more details put in. For example when the authors comment on the "direct probing of the fundamental absorption bands of nearly all gas molecules" please give a specific example the average reader can understand. This would help set the context for this paper and at the same time, the authors can discuss the limitation we currently have from not having good detectors in this regime.

- The paragraph that follows this (starting "Another intense area of research") is not linked

well to the previous one. In fact the first three paragraphs (excluding the bolded one) do not flow well together and seem more like a collection of ideas. Integrating them together is essential for a smooth flowing story.

- On page 3 the authors say "One approach to increase sensitivity is to use Superconducting Nanowire Single-Photon Detectors (SNSPDs), which offer single-photon sensitivity from visible to MIR wavelengths, low dark counts, short recovery times, and low timing jitter. Recently, it was suggested to extend the sensitivity of those detectors into the MIR wavelength range by reducing the nanowire width to 30 nm. However, in the MIR the performance of SNSPDs decrease drastically in terms of detection efficiency and recovery time, even when operated at the required few degrees Kelvin." This is a very important paragraph but I would like to see it explained. How bad is the performance going to be at 3.1 μm for instance. Will they be better than the experimental results presented here?

- Next the authors say "This scenario changed when DTU Fotonik demonstrated that the efficiency of the up-conversion process can be dramatically enhanced by placing a nonlinear crystal inside a resonant cavity. A measured quantum efficiency of 20% has been achieved, due to the high circulating power within the cavity resulting from the low-loss cavity design. Here, we couple the up-converter module with a state-of-the-art silicon Single Photon Avalanche Photodiode (SPAD) to demonstrate single-photon counting in the MIR for quantum optics applications". This really makes it sound like the big innovation was from DTU Fotonik and that this work is just a logical extension of it. This seems to reduce the significance and impact of this work and its justification for publication in Nature Communications.

- How efficient are the high-efficient up-converter modules? Are they only 20%. If so, are they really high efficiency?

- Losses are quoted at 25 dB from the output facet of the SPDC crystal to the input of the single-photon detectors. This seems very high when working at the single photon level. Please discuss in more detail and the prospect for future improvements

- In table 1, quantum efficiency are given. The majority of the single photon detection community have moved from quantum efficiency to system detection efficiency. Please use the number for this instead. Next when a quantum efficiency for the module + SPAD is given at $10\% \times 65\% = 6.5\%$ it seems that no account of coupling efficiency between the module and SPAD is considered. It must be included for a fair comparison.

Now let me summarize my opinion. I believe this is an interesting article, however from what is presented, it is not obvious it gives a better performance than other approaches. The authors need to make sure such a comparison is done in a fair way. Given this I can not recommend it for publication in its current form. A major revision is needed.

Q1:

I find the first bold text paragraph quite terse and not generally helpful introducing what they are doing, why it is important and its implications for the future. I believe this paragraph needs to be rewritten. Next in this paragraph, the authors say "Most quantum sensing devices use visible light". I do not believe this is true any more - especially for magnetic field sensors. Many of these operate coming outside the visible light spectrum. Further the authors also say "then detected with low-noise silicon avalanche photodiodes with single-photon counting ability and without the necessity of any cooling system". While they may not have chosen to use a cooling system, they are more efficient detectors at this wavelength than silicon avalanche photodiodes that do require active cooling. Use better and more efficient detectors is likely to be an advantage for many quantum applications.

A1:

We thank the reviewer for the comment. We change the first bold text paragraph accordingly, by addressing all the points raised by the reviewer (please, see the revised manuscript).

Q2:

In the next paragraph "the mid-infrared (MIR) spectral region (2-25 μm) is of interest to many areas of science and technology for mainly three reasons ..." the authors give a short explanation as to why single photon detection is important here. I like this as it helps set the context, but I want to see it explained and more details put in. For example when the authors comment on the "direct probing of the fundamental absorption bands of nearly all gas molecules" please give a specific example the average reader can understand. This would help set the context for this paper and at the same time, the authors can discuss the limitation we currently have from not having good detectors in this regime.

A2:

We thank the reviewer for this pertinent comment. We share his/her opinion that more scientific hints, would help to better set the context of the paper. Please, see the revised manuscript for the changes made to the main text.

Q3:

The paragraph that follows this (starting "Another intense area of research") is not linked well to the previous one. In fact the first three paragraphs (excluding the bolded one) do not flow well together and seem more like a collection of ideas. Integrating them together is essential for a smooth flowing story.

A3:

We rewrite the first three paragraphs taking into account the reviewer's suggestions (please, see the revised manuscript).

Q4:

On page 3 the authors say "One approach to increase sensitivity is to use Superconducting Nanowire Single-Photon Detectors (SNSPDs), which offer single-photon sensitivity from visible to MIR wavelengths, low dark counts, short recovery times, and low timing jitter. Recently, it was suggested to extend the sensitivity of those detectors into the MIR wavelength range by reducing the nanowire width to 30 nm. However, in the MIR the performance of SNSPDs decrease drastically in terms of detection efficiency and recovery time, even when operated at the required few degrees Kelvin." This is a very important paragraph but I would like to see it explained. How bad is the performance going to be at 3.1 μm for instance. Will they be better than the experimental results presented here?

A4:

We share the reviewer opinion that this is a very important paragraph, since it actually introduces the SNSPDs technology to perform single-photon counting. Let us comment the sensitivity of the SNSPDs moving from telecom wavelength to 3.1 μm . At around 1.55 μm in 2013 it has been published an astonishing paper on Nature Photonics, entitled "Detecting

single infrared photons with 93% system efficiency” (reference [25] of the revised manuscript). Here the authors reported 90% detection efficiency in the range 1.52-1.62 μm , a device dark count rate (measured with the device shielded from any background radiation) of 1 cps, timing jitter of ≈ 150 ps, full-width at half-maximum(FWHM) and reset time of 40 ns. Please note, that the system dark count rate is, instead, of about 1000 cps. This is dominated by background photons. Now, if one moves to 3.1 μm , the SNSPDs performances decrease drastically. Up to our knowledge, we found only one reference work which deals with 30-nm-wide SNSPDs in the MIR, by Marsili et. al (2012), ref [15] of the manuscript. In this work the authors claim a system detection efficiency of about 2% up to 5 μm , 10 ns of reset time and dark count rate of the order of 100 cps (measured with a metal shutter at $T \sim 1.5$ K). The dark count rate, instead, increases to a value larger than 10 kcps when the optical fibers, which deliver the optical signals to the SNSPDs, are coupled to the device. This is because, even in absence of any optical signal, the fibers are source of black body radiation being at room temperature. As it is discussed in detail in table 1, we believe that the performances of our method outperformed the ones of the SNSPDs operated at around 3.1 μm . Moreover, to the best of our knowledge, no further work has followed ref [15], in the MIR spectral region.

Q5:

Next the authors say "This scenario changed when DTU Fotonik demonstrated that the efficiency of the up-conversion process can be dramatically enhanced by placing a nonlinear crystal inside a resonant cavity. A measured quantum efficiency of 20% has been achieved, due to the high circulating power within the cavity resulting from the low-loss cavity design. Here, we couple the up-converter module with a state-of-the-art silicon Single Photon Avalanche Photodiode (SPAD) to demonstrate single-photon counting in the MIR for quantum optics applications". This really makes it sound like the big innovation was from DTU Fotonik and that this work is just a logical extension of it. This seems to reduce the significance and impact of this work and its justification for publication in Nature Communications.

A5:

trivial Yes, our work is based on the significant achievement reported in Nature Photonics [28] by the DTU group. However, it is not a simple logical extension and it is far from being a trivial experiment, since it required significant improvements with respect to the work reported previously. Specifically, a precise optimization of the conversion efficiency, process bandwidth and collected noise. In particular, the wide bandwidth SPDC process prevents the use of narrow band filter on the up-converted radiation (as done in [28], because of the narrow band input laser radiation source); therefore, the photon counter gathers a larger amount of noise. As reported in [28], the noise has different origins such as the up-converted black body radiation of the oven, where the nonlinear crystal is placed, and spurious parametric processes originated by poling errors. The noise gathered by the photon counter area can be so high to prevent the measure of any coincidence. Since this noise happens in the same spectral region of the SPDC generated radiation, it is actually impossible to filter it out through a spectral selection method. However, the noise is generated at several angles with respect to the crystal axis, thus can be partially filtered by means of a pinhole placed after the output lens of the up-converter module. The pinhole allows to select the best compromise between the system detection efficiency and the background count rate. Moreover, the performances reported in [28] are “ideal performances” for applications in high sensitivity spectral imaging, rather than for a real single-photon level experiment. In

our manuscript, we face, instead, a real single photon counting problem, such as a coincidence experiment, using 2 up-converter modules.

We were able, for the first time to the best of our knowledge, to demonstrate that it is possible to use this kind of technology for quantum optics experiments. We are also paving the way for correlation experiments suggesting a proper set-up that exploits the benefits that the up-conversion technology can offer.

Q6:

How efficient are the high-efficient up-converter modules? Are they only 20%. If so, are they really high efficiency?

A6:

We thank the reviewer for the question, which actually allows us to better discuss the efficiency of the up-converter modules exploited in this work. Other frequency up-conversion schemes that have been reported so far, exhibited typical sum frequency generation efficiencies of the order of 10^{-5} [29], which are actually orders of magnitude lower with respect to the value of 20% reported in ref [28].

So the answer is “yes”, they are really high efficient due to the resonant cavity scheme.

Q7:

Losses are quoted at 25 dB from the output facet of the SPDC crystal to the input of the single-photon detectors. This seems very high when working at the single photon level. Please discuss in more detail and the prospect for future improvements

A7:

We thank again the reviewer for the pertinent comment. We agree with the reviewer that the total losses seems to be quite high. Actually, we can distinguish between intrinsic losses of the detection method and losses that can be quite easily removed from the total 25 dB.

As it is reported in the methods section, 10 dB comes from the 10% quantum efficiency of the up-conversion process, which can be enhanced to 20% [28]. For the remaining 15 dB, 14 dB comes from a non-optimized overlap of the up-converted visible photon waist at the small 180 μm SPAD (SPCM-AQRH-14) active area; the remaining 1 dB is half due to the transmittance of the filter (see Supplementary information), while the other contribution comes from the CaF_2 lens (mounted before the up-conversion stage). To improve the detection method, we can consider, instead, a 20% of up-conversion efficiency and 80% of quantum efficiency of the best silicon-based photon counter up to our knowledge (ID 120) [44] which has also the advantage to have 500 μm of active area. The use of this SPAD, together with an optimized design of the up-converted visible photon waist at the detector, will allow reducing the 14 dB of coupling losses and, eventually, totally eliminating them.

Therefore in the optimized system, the limiting intrinsic losses will be about 8.5 dB ($20\% \cdot 80\% \cdot 90\%$), which corresponds to a System Detection Efficiency up to almost 15%.

Still we think that the current experiment is a useful proof of concept demonstration showing that, with our method, it is indeed possible to perform MIR quantum optics experiments, even though we are not working in an optimal configuration. Moreover, our experiment gives also an idea of the strength of the method, since we were able to perform single-photon measurements in the MIR, even though we faced not negligible coupling

losses.

Furthermore, we are currently working to couple either the input and the output of the up-converter module, with optical fibers. This will significantly help the ease of input-output manipulation of the light with respect to the module, as well as, at the same time, drastically reduce the coupling losses with the detector.

Q8:

In table 1, quantum efficiency are given. The majority of the single photon detection community have moved from quantum efficiency to system detection efficiency. Please use the number for this instead. Next when a quantum efficiency for the module + SPAD is given at $10\% \times 65\% = 6.5\%$ it seems that no account of coupling efficiency between the module and SPAD is considered. It must be included for a fair comparison.

A8:

We thank the reviewer for the pertinent comment. He/She correctly stated that in Table 1 the coupling losses between the up-converter module and the SPAD are not taken into account. Indeed in the table is reported the quantum efficiency and not the system detection efficiency. We updated the table by adding the system detection efficiency entry and providing a detailed discussion about the removable optical coupling losses (please, see the updated Table 1 in the revised manuscript).

Reviewer #2 (Remarks to the Author):

This paper is a useful demonstration showing upconversion of MIR twin photons to a wavelength where they can be detected by a silicon photon counter but there are a number of issues that need addressing before publication, most of which have to do with clarity.

Table 1 claims that the overall QE for the detection is $6.5\% = 10\%$ upconversion \times 65% SPAD but Methods section says “the total loss from the output facet of the SPDC PPLN to the single photon detector has been estimated to be 25 dB. 10 % comes from the conversion efficiency of both upconverter modules which is 10%.” Why is the additional 15 dB not part of the efficiency of the scheme? Something is inconsistent.

The units of most of the count rates mentioned should be “counts per second” not “Hz” which is defined as cycles/second.

In the conclusion it is claimed that a measurement of entangled MIR photons was presented. What was presented was a measurement of correlated photons, not entangled photons. No entanglement was demonstrated.

Reference is made to the “dark count rate in both upconverter modules”. What is being referred to is really the “background rate” which is due to an optical process. This is not what is usually meant by the term “dark counts”.

A minor point, but it is not quite correct to say that the scheme operates at room temperature, as the SPADs are TE cooled.

In the methods section, it mentions that the PPLN has a number of waveguides with different poling periods, but it is never stated (at least until much later) that just one of those is chosen for the experiment. For clarity, that should be stated explicitly. A simple additional phrase would fix this.

“A number of filters were included to remove unwanted light from entering the SPADs.” Filtering in the presence of strong pump light is key to the success of this scheme, so it would be good specifically list the transmittances of those filters.

Fig. 2 caption: It states that “the bandwidth is reported also in the graph.” No it is not, but it should be. Also I would use arrows to point from the labels to the curves. As it is, it can be hard to figure out which color is which.

In the supplementary section Fig. S1 has no color scale bar.

In the supplementary section it states “A trade-off has been found in term of signal to noise ratio, at a current value of 3.6 A.” It would be useful to show a graph of that.

Figure S3 is really a trivial result that the does not warrant an additional graph especially when Fig. 3b already shows it.

The last page of the supplementary section essentially presents how the efficiencies of each of the two detection channels can be calibrated by a two photon source. This calibration technique has been demonstrated numerous times. There should at least be some reference included to provide the reader a clue to that.

Q1:

Table 1 claims that the overall QE for the detection is $6.5\% = 10\%$ upconversion x 65% SPAD but Methods section says “the total loss from the output facet of the SPDC PPLN to the single photon detector has been estimated to be 25 dB. 10 % comes from the conversion efficiency of both upconverter modules which is 10%.” Why is the additional 15 dB not part of the efficiency of the scheme? Something is inconsistent.

A1:

We apologize for not being clear in the first version of the manuscript. The answer to this question is discussed in A7 and A8 for Reviewer #1. Please, see the updated Table 1 in the revised manuscript.

Q2:

The units of most of the count rates mentioned should be “counts per second” not “Hz” which is defined as cycles/second.

A2:

We thank the reviewer for this pertinent comment. We modified the text of the manuscript accordingly.

Q3:

In the conclusion it is claimed that a measurement of entangled MIR photons was presented. What was presented was a measurement of correlated photons, not entangled photons. No entanglement was demonstrated.

A3:

We totally agree with the reviewer. The coincidence measurement demonstrate that we collected correlated photon pairs, but it is not sufficient to claim any kind of entanglement. We modify the text of the manuscript accordingly.

Q4:

Reference is made to the “dark count rate in both upconverter modules”. What is being referred to is really the “background rate” which is due to an optical process. This is not what is usually meant by the term “dark counts”.

A4:

We agree with the reviewer that what we referred to is the “background rate” due to optical processes and not the dark counts. We modify the text of the manuscript accordingly.

Q5:

A minor point, but it is not quite correct to say that the scheme operates at room temperature, as the SPADs are TE cooled.

A5:

We thank the reviewer for the pertinent comment. He/She correctly states that our SPADs are thermoelectrically (TE) cooled. We modify the text of the manuscript accordingly.

Q6:

In the methods section, it mentions that the PPLN has a number of waveguides with different poling periods, but it is never stated (at least until much later) that just one of those is chosen for the experiment. For clarity, that should be stated explicitly. A simple additional phrase would fix this.

A6:

This is obvious. However since it was raised, we modify the text of the manuscript accordingly.

Q7:

“A number of filters were included to remove unwanted light from entering the SPADs.” Filtering in the presence of strong pump light is key to the success of this scheme, so it would be good specifically list the transmittances of those filters.

A7:

We add the filter transmittance spectrum in the range of interest (700-900 nm) for each module in the revised Supplementary Information section, Fig. S4.

Q8:

Fig. 2 caption: It states that “the bandwidth is reported also in the graph.” No it is not, but it should be. Also I would use arrows to point from the labels to the curves. As it is, it can be hard to figure out which color is which.

A8:

We modify Fig. 2b following the reviewer’s suggestion.

Q9:

In the supplementary section Fig. S1 has no color scale bar.

A9:

We apologize for this. We add the color scale bar to Fig. S1. In the revised manuscript the old Fig. S1a is now part of the new Fig. 3a, as explicitly requested by Reviewer #3 in Q1.

Q10:

In the supplementary section it states “A trade-off has been found in term of signal to noise ratio, at a current value of 3.6 A.” It would be useful to show a graph of that.

A10:

Following the reviewer’s suggestion, we add a graph of the Coincidence to Accidental Ratio (CAR) as a function of the injected current in the Supplementary Information section in Fig. S3.

Q11:

Figure S3 is really a trivial result that the does not warrant an additional graph especially when Fig. 3b already shows it.

A11:

The old Figure S3 is no longer present in the revised Supplementary Information section.

Q12:

The last page of the supplementary section essentially presents how the efficiencies of each of the two detection channels can be calibrated by a two photon source. This calibration technique has been demonstrated numerous times. There should at least be some reference included to provide the reader a clue to that.

A12:

Two references have been added to the simulated coincidence measurement method.

Reviewer #3 (Remarks to the Author):

The authors extend the method of spectral conversion by sum-frequency generation in a nonlinear crystal to the MIR spectral region, which is of high relevance to science and applications. The idea was demonstrated previously in different spectral regions. Still, in my opinion, the current demonstration is relevant technologically.

The manuscript is well-written. The methods section is detailed, which is important for a technical contribution.

It would be interesting to see more experimental results in the paper. For example a part of Fig. S1 could be added as an inset to Fig. 1.

An exemplary measurement (single photon interference or Hong-Ou-Mandel effect) performed with the help of the presented detection scheme would add value to the manuscript. Alternatively the authors could demonstrate the filtering capabilities of the scheme and measure the spectrum of SPDC using the relation presented in Fig. S1 c)

Let me go over the text giving more specific remarks. Numbers given enumerate the lines of the manuscript.

75. I understand that it might be advantageous to have spatial and spectral filtering, but it is often a drawback that a detector has a limited bandwidth. We see this problem even in the reported situation (see Fig. 2 b). I suggest to reformulate this part.

88-91. What authors write here is a simplification, which is in general not true. Signal and idler photons don't need to be correlated -- this depends on the parameters of the pump, crystal and collection optics.

92. Formula 1: The authors discuss quasi phase matching (QPM), but use a formula without the contribution proportional to $1/(\text{poling period})$ which is the key point for QPM. This should be corrected.

104. Minor comment: Just to be precise and clear I wouldn't say that the spectrum is symmetrically distributed around λ_{deg} (this would be true for ω_{deg} ...).

175. Which time slot do the authors refer to here? In my understanding time-slot relevant to CAR is the coincidence window width and the time relevant to multi-pair emission is the coherence time of SPDC. This should be clarified.

Additionally, saturation of the detectors could be mentioned as another limitation (I realize that it is due to APDs and not to the conversion, but for the completeness it might be mentioned).

268. Fig. 1. a) could be more schematic -- simplified to increase clarity. d) it is not written in the inset how the data for this plot was measured.

333. Table 1: It would be informative to add the dark count rate of SNSPDs @ 1.5K.

In summary, I think that the manuscript could be accepted after some revisions .

Q1:

It would interesting to see more experimental results in the paper. For example a part of Fig. S1 could be added as an inset to Fig. 1.

An exemplary measurement (single photon interference or Hong-Ou-Mandel effect) performed with the help of the presented detection scheme would add value to the manuscript. Alternatively the authors could demonstrate the filtering capabilities of the scheme and measure the spectrum of SPDC using the relation presented in Fig. S1 c)

A1:

We agree with the reviewer that Fig. S1 can be added in the main text to show in the paper more experimental results. We actually add another figure in the revised manuscript (Fig. 3), where we show the experimental characterization previously reported in Fig. S1a. In Fig. 3b an experimental demonstration of the filtering capability of the method at $3.1 \mu\text{m}$ is reported. In Fig. 3c, the measured SPDC spectrum is reported. Fig. 3c is obtained by varying both the SPDC crystal temperature, the poling period and the temperature of the PPLN crystal inside the up-converter module.

Q2:

75. I understand that it might be advantageous to have spatial and spectral filtering, but it is often a drawback that a detector has a limited bandwidth. We see this problem even in the reported situation (see Fig. 2 b). I suggest to reformulate this part.

A2:

We thank the reviewer for the pertinent comment. We agree that having spatial and spectral filtering can have drawbacks, such as not being able at a glance to collect all the SPDC spectrum, as it is shown in the old Fig. 2b. We reformulate this part in the main text, taking into account the reviewer's suggestion.

Q3:

88-91. What authors write here is a simplification, which is in general not true. Signal and idelr photons don't need to be correlated -- this depends on the parameters of the pump, crystal and collection optics.

A3:

We thank the reviewer for the pertinent comment. We totally agree that the degree of correlation of a photon pair source in general depends on the parameters of the pump, the crystal and the collection optics. We reformulate this part in the main text accordingly.

Q4:

92. Formula 1: The authors discuss quasi phase matching (QPM), but use a formula without the contribution proportional to $1/(\text{poling period})$ which is the key point for QPM. This

should be corrected.

A4:

We apologize for this omission. We thank the reviewer for the pertinent comment. He/She correctly stated that the quasi phase matching term is not present in the momentum conservation relation, which is Eq. 1. We agree that the formula is more clear if the $1/(\text{poling period})$ contribution manifestly appears in eq 1. We modify Eq. 1 accordingly.

Q5:

104. Minor comment: Just to be precise and clear I wouldn't say that the spectrum is symmetrically distributed around λ_{deg} (this would be true for ω_{deg} ...).

A5:

We totally agree with the reviewer that the SPDC spectrum is symmetrically distributed in energy around the degenerate frequency $\omega_{\text{deg}} = (\omega_{\text{pump}})/2$, where ω_{pump} is the pump frequency. The SPDC spectrum, instead, is not symmetrically distributed in wavelength around the degenerate wavelength λ_{deg} . For the sake of clarity, we fix it in the main text with an additional phrase.

Q6:

175. Which time slot do the authors refer to here? In my understanding time-slot relevant to CAR is the coincidence window width and the time relevant to multi-pair emission is the coherence time of SPDC. This should be clarified.

A6:

CAR is calculated taking the number of coincidences within the coincidence window of 1.33 ns, over the average of the background on the same time window taken apart from the peak. While, instead, the bandwidth of the up-converted signal and idler photons corresponds to a photon coherence time of \square ps, much lower than the coincidence window. Therefore, the width of the coincidence peak is given by the response time of the coincidence window.

Q7:

Additionally, saturation of the detectors could be mentioned as another limitation (I realize that it is due to APDs and not to the conversion, but for the completeness it might be mentioned).

A7:

We agree with the reviewer that saturation count rate of the APDs is an important parameter that deserves to be mentioned. As it is a technical detail, we wrote an additional phrase at the end, in the methods section to fix it.

Q8:

268. Fig. 1. a) could be more schematic -- simplified to increase clarity. d) it is not written in

the inset how the data for this plot was measured.

A8:

Fig. 1a has been modified in order to increase clarity. Fig. 1d is now in the new Fig. 3c of the revised manuscript. A detailed comment of the experimental measurement is discussed in the main text (please, see the revised text).

Q9:

333. Table 1: It would be informative to add the dark count rate of SNSPDs @ 1.5K.

A9:

In the updated Table 1 of the revised manuscript, there is also the information about the dark count rate of SNSPDs @ 1.5K.

REVIEWERS' COMMENTS:

Reviewer #1 (Remarks to the Author):

Dear Editor,

The authors have addressed my queries from the previous report. I am happy to suggest it be accepted.

Reviewer #2 (Remarks to the Author):

Significant improvements have been made, but there are still issues to be fixed before publishing. This issues are all fixable and not show stoppers.

The English still needs some work. To facilitate that, I have included some specific rewrite suggestions below along with the technical issues to be addressed.

Line 39 "allows experiments in"

44 "One of the key experiments"

45 "measurement which relies"

142 How about using a more common color like purple instead of an obscure color like fuchsia?

Fig 2b The y axis should be labeled normalized not arbitrary units.

195 should read: "counts from uncorrelated photons that happen to arrive at the same time as the coincidence photons and true coincidence counts are indistinguishable,"

259 Why is Coverision.com being referenced for the optimum Rayleigh range? That must be a typo.

282 Should read: "maximum count rate before 100% saturation"

This would make it clear that you would never run anywhere close to that maximum rate

285 This line says that the total loss is 25 dB and then contributions are listed as:

10 dB for conversion efficiency

0.45 dB for lens 1

0.5 dB for filters

Then it says "The remaining losses are due to a non-perfect overlap with the active area of the silicon detector, which has 180 μm diameter."

The loss for this component should be given in dB too, so it is clear where that all the component losses add up.

Also if the beam radius is 97 microns and the detector radius is 90 microns why is the loss 10dB? That seems way too high. And then there is the question of why a tighter focus was not used? That should be explained. The paragraph added after line 294 suggests using a larger detector in the future, but does not address why a tighter focus was not used. Certainly, the spot could have been reduced by a factor of 2.

I suggest that for clarity, that these component efficiencies be specifically listed as additional rows in

Table 1. Component efficiencies for the SNSPDs could be given in the table also. That would provide the clearest comparison of the technique.

line 287. Also I think the lens referred to with the 0.45 dB should be L2 not L1

Figure 3 Caption need to state that the color bar is efficiency.

Fig 3b does not show the filtering capability of the module. It shows the temperature sensitivity. A vertical slice shows the filtering capability.

Last line about line 364 the English garbled. It should read something like this:

"Since black body radiation is a main source of noise in the MIR, we believe the reported value without the cold shutter is the fairest value for this comparison."

Supplementary

24 "Moreover, in Fig. S1(a,b), the bandwidths for the 5 mm long and 20 mm long crystals for a poling 24 period of 21.5 μm , are reported for comparison"
should read

"Moreover, in Fig. S1(a,b), the bandwidths for the 5 mm long and 20 mm long crystals for a poling 24 period of 21.5 μm , the differences in the bandwidths can be seen."

Fig S2 the order of the legend should match the order of the curves. That is, the legend should list 4 A on the top and 1 A on the bottom, just like the data.

33 BAL should be defined here. It is defined in the main paper, but it should be included here too.

43 An explanation is given for the decrease in CAR as the laser pump current, and thus the conversion efficiency, increases. It is stated that as the efficiency increases they start to see background thermal radiation being upconverted. The problem is that as the efficiency rises, the signals from both the desired photon and the thermal background should rise together, ultimately reaching background limit performance. That limit should show saturation, not an ultimate decrease. So some other explanation is needed to account for the rolloff.

94 & 109 Entangled is still used. It should be replaced by "correlated"

Reviewer #3 (Remarks to the Author):

The points that I raised have been answered satisfactorily. I have no further comments. I think that the manuscript could be accepted in the current form.

Point-by-point response to Reviewer #2

Q1:

The English still needs some work. To facilitate that, I have included some specific rewrite suggestions below along with the technical issues to be addressed.

(...)

A1:

We thank the reviewer for the comment. We modified the English following his/her suggestions in the final revision of the manuscript.

Q2:

142 How about using a more common color like purple instead of an obscure color like fuchsia?

Fig 2b The y axis should be labeled normalized not arbitrary units.

A2:

In Figure 2 we have now opted to use a green color, in place of fuchsia. In this way, we hope the figure is actually more accessible. In panel b of Figure 2 we also labeled the y axis "Normalized Efficiency".

Q3:

195 should read: "counts from uncorrelated photons that happen to arrive at the same time as the coincidence photons and true coincidence counts are indistinguishable,"

A3:

We modify the text accordingly.

Q4:

259 Why is Coversion.com being referenced for the optimum Rayleigh range? That must be a typo.

A4:

We thank the reviewer for the comment. Actually the Coversion company, where we bought the Periodically Poled Lithium Niobate (PPLN) nonlinear SPDC crystal, provides a support guide to the use of the PPLN. They suggest the following:

"In general, a good rule of thumb is that the spot size should be chosen such that the Rayleigh range is half the length of the crystal. The spot size can then be reduced in small increments until the maximum efficiency is obtained. "

We have experimentally verified the focusing condition in our proposed experimental implementation by using different lenses, which actually determine different beam waist and Rayleigh range within the SPDC crystal (see Figure 1c). We found that the optimum condition in term of SPDC generation efficiency was achieved when the Rayleigh range is approximately half the length of the crystal, as it was written in the manuscript. Please note that before this experiment, we also gained expertise in efficient SPDC generation from PPLN from the recently published work “Trenti, A., et al. "Quantum interference in an asymmetric Mach-Zehnder interferometer." *Journal of Optics* 18.8 (2016).” Therefore, the reference to covesion.com is not really a typo, but since it was considered misleading we removed it, without any loss of information.

Q5:

282 Should read: “maximum count rate before 100% saturation”
This would make it clear that you would never run anywhere close to that maximum rate

A5:

We thank the reviewer for this comment. For a free running SPAD, the maximum count rate estimate is given by the inverse of the dead time (20 ns in our case). Generally, it is recommended to not use the SPAD near 100% saturation, because the detector start to respond in a nonlinear fashion with respect to the incident photon flux. However, also in this case, the true photon flux incidents upon the detector can be derived, by statistically taking into account that the more the incident photons, the more the probability that some photon detections can be lost due to the fact the detector gate is increasingly closed. In any case, it is more comfortable to use the detector at an average count rate far from the saturation, as it was our case $150 \text{ kHz} \ll 40 \text{ Mhz}$, where you do not have to worry about nonlinear correction.

Q6:

285 This line says that the total loss is 25 dB and then contributions are listed as:
10 dB for conversion efficiency
0.45 dB for lens 1
0.5 dB for filters
Then it says “The remaining losses are due to a non-perfect overlap with the active area of the silicon detector, which has 180 μm diameter.”
The loss for this component should be given in dB too, so it is clear where that all the component losses add up.
Also if the beam radius is 97 microns and the detector radius is 90 microns why is the loss 10dB? That seems way too high. And then there is the question of why a tighter focus was not used? That should be explained. The paragraph added after line 294 suggests using a larger detector in the future, but does not address why a tighter focus was not used.
Certainly, the spot could have been reduced by a factor of 2.

I suggest that for clarity, that these component efficiencies be specifically listed as additional rows in Table 1. Component efficiencies for the SNSPDs could be given in the table also. That would provide the clearest comparison of the technique.

A6:

We thank the reviewer for the pertinent comment, which actually gives us the possibility to explain a little bit more in details the alignment procedure and related issues.

First, let us list explicitly the various losses contribution that add up to the final value of 25 dB. 25 dB accounts for the total loss from the output facet of the SPDC PPLN to the single-photon detector. The various contributions are outlined in the following:

-) 10 dB comes from the module up-conversion efficiency (10% of efficiency);
-) 0.45 dB come from the transmission losses of L2;
-) 0.5 dB comes from the transmission losses of the filtering stage between the module and the SPAD;
-) 1.9 dB comes from the SPAD detection efficiency (65% in dB);
-) The remaining $25 - (10+0.45+0.5+1.9) = 12.15$ dB are due to a non-optimized overlap between the up-converted photons and the SPAD circular active area.

We totally agree with the referee that 12.15 dB are too high considering 97 μm of beam radius over the detector radius of 90 μm . Indeed, from Gaussian optics theory the beam intensity is expected to decrease by the factor $1/e^2 \approx 0.135$ at the radial distance of the beam radius. Considering as a first approximation to make the calculation easy the beam radius to be equal to the detector one, this means that about 86% of the power can be detected (about 0.7 dB). Actually, this is an underestimation, because the simulated beam radius is slightly higher with respect to the detector radius and we expect that the efficiency of the circular active area decreases at the edge. Nevertheless, we think that it fixes the order of magnitude of the expected coupling losses, which are significantly lower with respect to the measured 12.15 dB value. It is possible to estimate the beam radius, by varying the detector position within the plane of the beam. With this procedure, the beam-detector overlap is mapped for different relative position and the effective beam radius can be derived. The effective beam radius at the detector area was measured to be 300 μm . Now comes the question about the tighter focus. We totally agree with the referee that a tighter focus could have reduced the coupling losses, as it decreases the spot size at the detector. Actually, we tried to use lenses with shorter focal lengths in place of L3, whose focal length is 7.5 cm. However, we were not able to increase the signal to noise ratio. There are some technical issues that led this result. If on one hand a tighter focus decreases the up-converted spot size, on the other hand it also reduces the Rayleigh range yielding the alignment procedure more and more sensitive. To better understand this, it is worthy to explain the module-SPAD alignment procedure. As a first step the SPAD is aligned with respect to the up-converter module thanks to the Second Harmonic Generation in the PPLN crystal of the 1064 nm pump, at 532 nm of the module (see Supplementary Figure 2). Then, as a second step, the detector alignment is optimized by maximizing the counts of the SPDC up-converted photons (at this point, the filter stage is placed after the module). We experienced that the alignment of the 532 nm signal was slightly different with respect to the up-converted radiation, due to chromatic aberration of the lens L3. This issue is amplified by using shorter focal lengths, as the alignment is more sensitive in that case. Moreover, spherical aberrations are also present, and they are amplified by shorter focal lengths as well.

In light of the reasons explained above, we think that in order to reduce the optical coupling losses, it would be easier the use of a larger area detector instead of a tighter focus.

At the same time, we are currently working to couple both the input and the output of the up-converter module, with optical fibers. This will significantly help the ease of input-output manipulation of the light with respect to the module, as well as, at the same time, drastically reduce the coupling losses with the detector.

Following the referee suggestion, the component efficiencies are specifically listed as an additional entry in Table 1 of the revised manuscript.

Q6:

line 287. Also I think the lens referred to with the 0.45 dB should be L2 not L1

A6:

We thank the reviewer for the pertinent comment. He/She correctly stated that the lens referred to with 0.45 dB is L2 and not L1 (see Fig. 1a). This was a typo and we apologize for it.

Q7:

Figure 3 Caption need to state that the color bar is efficiency.
Fig 3b does not show the filtering capability of the module. It shows the temperature sensitivity. A vertical slice shows the filtering capability.

A7:

We thank the reviewer for the pertinent comment. We add in the caption of Fig. 3 that the color bar is the efficiency normalized for the maximum value. The old Fig. 3b shows the module response when it is set fixed to phase match at 3.1 μm (21.5 μm poling period, and 145°C crystal temperature), as a function of the SPDC crystal temperature. As it can be appreciated from Fig. 2b, the SPDC emitted spectrum significantly changes from the degenerate emission, which occurs at 135°C of the SPDC crystal, to a more and more non-degenerate emission as the temperature is shifted from that value. The module actually filter the incoming SPDC radiation as a narrow bandpass filter centered at 3.1 μm with a bandwidth of 7 nm. Following the referee suggestion, we add a new panel in Fig. 3, where the new Fig. 3b is a vertical slice of Fig. 3a for the 21.5 μm poling period, 145°C crystal temperature. Fig. 3b clearly shows a characterization of the module filtering capability at 3.1 μm .

Q8:

Last line about line 364 the English garbled. It should read something like this:
“Since black body radiation is a main source of noise in the MIR, we believe the reported value without the cold shutter is the fairest value for this comparison.”

A8:

We modify the text accordingly.

Q9:

Supplementary

24 “Moreover, in Fig. S1(a,b), the bandwidths for the 5 mm long and 20 mm long crystals for a poling 24 period of 21.5 μm , are reported for comparison” should read

“Moreover, in Fig. S1(a,b), the bandwidths for the 5 mm long and 20 mm long crystals for a poling 24 period of 21.5 μm , the differences in the bandwidths can be seen.”

A9:

We modify the text of the Supplementary accordingly.

Q10:

Fig S2 the order of the legend should match the order of the curves. That is, the legend should list 4 A on the top and 1 A on the bottom, just like the data.

A10:

In the revised Supplementary, the legend list 4 A on the top and 1A on the bottom as the data.

Q11:

33 BAL should be defined here. It is defined in the main paper, but it should be included here too.

A11:

In the revised Supplementary, the BAL acronym is written in full.

Q12:

43 An explanation is given for the decrease in CAR as the laser pump current, and thus the conversion efficiency, increases. It is stated that as the efficiency increases they start to see background thermal radiation being upconverted. The problem is that as the efficiency rises, the signals from both the desired photon and the thermal background should rise together, ultimately reaching background limit performance. That limit should show saturation, not an ultimate decrease. So some other explanation is needed to account for the rolloff.

A12:

We thank the reviewer for the pertinent comment. We agree with the referee that by increasing the laser pump current, ultimately a saturation would be expected in terms of signal to noise ratio. The reason for the CAR dropping in Supplementary Figure 3, is related to high-power effects within the up-converter module cavity above 3.6 A.

These effects are expected to come into play due to the high intra-cavity CW circulating power (100 W at 3.6 A). We experimentally characterized that for current value higher than 3.6 A, the intra-cavity circulating power is not stable as a function of the time, as it is the case for current value until 3.6 A.

This is mainly due to high intensity effects, such as thermal lens, which actually changes significantly the beam waist, which in turn affects the module conversion efficiency.

Q13:

94 & 109 Entangled is still used. It should be replaced by “correlated”

A13:

We thank the reviewer for the pertinent comment. As it was discussed in the previous response letter, a coincidence measurement alone demonstrate the detection of correlated photon pairs, which it is not an entanglement demonstration. Therefore, again, we agree with the referee suggestion and “entangled” is replaced with “correlated”.